# Improving Explorability in Variational Inference with Annealed Variational Objectives

**Chin-Wei Huang**[†,⋆,1]  **Shawn Tan**[†,2]  **Alexandre Lacoste**[⋆,3]  **Aaron Courville**[†‖,4]
[†]MILA, University of Montreal    [⋆]Element AI    [‖]CIFAR Fellow
[1]chin-wei.huang@umontreal.ca, [2]jing.shan.shawn.tan@umontreal.ca
[3]allac@elementai.com, [4]aaron.courville@umontreal.ca

## Abstract

Despite the advances in the representational capacity of approximate distributions for variational inference, the optimization process can still limit the density that is ultimately learned. We demonstrate the drawbacks of biasing the true posterior to be unimodal, and introduce Annealed Variational Objectives (AVO) into the training of hierarchical variational methods. Inspired by Annealed Importance Sampling, the proposed method facilitates learning by incorporating energy tempering into the optimization objective. In our experiments, we demonstrate our method's robustness to deterministic warm up, and the benefits of encouraging exploration in the latent space.

## 1 Introduction

*Variational Inference* (VI) has played an important role in Bayesian model uncertainty calibration and in unsupervised representation learning. It is different from *Markov Chain Monte Carlo* (MCMC) methods, which rely on the Markov chain to reach an equilibrium; in VI one can easily draw i.i.d. samples from the variational distribution, and enjoy lower variance in inference. On the other hand VI is subject to bias on account of the introduction of the approximating variational distribution.

As pointed out by Turner and Sahani (2011), variational approximations tend not to propagate uncertainty well. This inaccuracy and overconfidence in inference can result in bias in statistics of certain features of the unobserved variable, such as marginal likelihood of data or the predictive posterior in the context of a Bayesian model. We argue that this is especially true in the amortized VI setups (Kingma and Welling, 2014; Rezende et al., 2014), where one seeks to learn the representations of the data in an efficient manner. We note that this sacrifices the chance of exploring different configurations of representation in inference, and can bias and hurt the training of the model.

This bias is believed to be caused by the variational family that is used, such as factorial Gaussian for computational tractability, which limits the *expressiveness* of the approximate posterior. In principle, this can be alleviated by having a more rich family of distributions, but in practice, it remains a challenging issue for optimization. To see this, we write the training objective of VI as the KL divergence, also known as the variational free energy $\mathcal{F}$, between the proposal $q$ and the target distribution $f$:

$$\mathcal{F}(q) = \mathbb{E}_q \left[ \log q(z) - \log f(z) \right] = \mathcal{D}_{\mathrm{KL}}(q||f).$$

Due to the KL divergence, $q$ gets penalized for allocating probability mass in regions where $f$ has low density. This behaviour of the objective can result in a cascade of consequences. The variational distribution becomes biased towards being excessively confident, which, in turn, can inhibit the variational approximation from escaping poor local minima, even when it has sufficient representational capacity to accurately represent the posterior. This usually happens when the target

(posterior) distribution has a complicated structure, such as multimodality, where the variational distribution might get stuck fitting only a subset of modes.

In what follows, we discuss two annealing techniques that are designed with two diverging goals in mind. On the one hand, *alpha-annealing* is used to encourage exploration of significant modes and is designed for learning a good inference model. On the other, *beta-annealing* facilitates the learning of a good generative model by reducing noise and regularization during training.

**Alpha-annealing:**   The optimization problem of inference is due to the non-convex nature of the objective, and can be mitigated via *energy tempering* (Katahira et al., 2008; Abrol et al., 2014; Mandt et al., 2016):

$$\mathbb{E}_q\left[\log q(z) - \alpha \log f(z)\right] \tag{1}$$

where $\alpha = \frac{1}{T}$ and $T$ is analogous to the temperature in statistical mechanics. The temperature is usually initially set to be high, and gradually annealed to 1, i.e. $\alpha$ goes from a small value to 1. The intuition is that when $\alpha$ is small the energy landscape is smoothed out, as $\nabla_z f(z)^\alpha = \alpha f(z)^{\alpha-1} \nabla_z f(z) = \alpha f(z)^\alpha \nabla_z \log f(z)$ goes to zero everywhere when $\alpha \to 0$ if $\log f$ is continuous and has bounded gradient.

However, alpha-annealing might not be ideal in practice. Tempering schemes are typically suitable for one time inference, e.g. in the case of inferring the posterior distribution of a Bayesian model, but it can be time consuming for latent variable models or hierarchical Bayesian models where multiple inferences are required for maximizing the marginal likelihood. Examples include deep latent variable models (Kingma and Welling, 2014; Rezende et al., 2014), filtering and smoothing in deep state space models (Chung et al., 2015; Fraccaro et al., 2016; Karl et al., 2016; Shabanian et al., 2017), and hierarchical Bayes with inference network (Edwards and Storkey, 2017). Doing energy tempering in these setups would do harm to the training due to the excess noise injected to the gradient estimate of the model.

**Beta-annealing:**   *Deterministic warm-up* (Raiko et al., 2007) is applied to improve training of a generative model, which is of even greater importance especially in the case of hierarchical models (Sønderby et al., 2016) and latent variable models with flexible likelihood (Gulrajani et al., 2017; Bowman et al., 2016). Let the joint likelihood of observed data $x$ and latent variable $z$ be $p(x, z) = p(x|z)p(z)$, which is equal to the true posterior $f(z) = p(z|x)$ up to a normalizing constant (marginal $p(x)$). The annealed variational objective (negative *Evidence Lower Bound*, or ELBO) is

$$\mathbb{E}_q[\beta(\log q(z) - \log p(z)) - \log p(x|z)] \tag{2}$$

where $\beta$ is annealed from 0 to 1. The rationale behind this annealing is that the first term in the parenthesis over-regularizes the model by forcing the approximate posterior $q(z)$ to be like the prior $p(z)$, and thus by reducing this *prior contrastive* coefficient early on during the training we allow the decoder to make better use of the latent code to represent the underlying structure of the data. However, the entropy term has less importance when the coefficient is smaller, leading it to be more a deterministic autoencoder and biasing the encoding distribution to be more sharp and unimodal. This approach is clearly in conflict with the principle of energy tempering, where one allows the approximate posterior to "explore" in the energy landscape for significant modes.

This work aims to clarify the implications of doing these two kinds of annealing, in terms of the inductive bias the learning objective and algorithm has. We first review a few techniques in the VI and MCMC literature to tackle the **expressiveness problem** (i.e. limited expressiveness of the approximate posterior) and the **optimization problem** (i.e. the mode seeking problem) in inference. We focus on the latter. These are summarized in Section 3. We introduce *Annealed Variational Objectives* (AVO) in Section 4, which aims to satisfy the criteria the alpha and beta annealing schemes seek to achieve separately. Finally, in Section 5 we demonstrate the biasing effect of VI with beta annealing, and demonstrate the relative robustness and effectiveness of the proposed method.

## 2   Related work

Naturally, recent works in VI have been focused on reducing representational bias, especially in the setting of amortized VI, known as the Variational Auto-Encoders (VAE) (Kingma and Welling, 2014;

Rezende et al., 2014), by (1) finding more expressive families of variational distributions without losing the computational tractability. Explicit density methods include Rezende and Mohamed (2015); Ranganath et al. (2016); Kingma et al. (2016); Tomczak and Welling (2016, 2017); Berg et al. (2018); Huang et al. (2018). Other methods include implicit density methods (Huszár, 2017; Mescheder et al., 2017; Shi et al., 2018). A second line of research focuses on (2) reducing the amortization error introduced by the use of a conditional network (Cremer et al., 2018; Krishnan et al., 2017; Marino et al., 2018; Kim et al., 2018).

In terms of non-parametric methods, the Importance Weighted Auto-Encoder (IWAE) developed by Burda et al. (2016) uses several samples for evaluating the loss to reduce the variational gap, which can be expensive in scenarios where the decoder is much more complex. Nowozin (2018) further reduces the bias in inference via *Jackknife Variational Inference*. However, Rainforth et al. (2018) notices that the signal-to-noise ratio of the encoder's gradient vanishes with increasing number of importance samples (reduced bias), rendering the encoder an ineffective representation model in the limit case. Salimans et al. (2015) combines MCMC with VI but the inference process requires multiple passes through the decoder as well.

Our method is orthogonal to all these, assuming we have a rich family of approximate posterior at hand, and smooths out the objective landscape using annealed objectives specifically for hierarichical VI methods. We also note that it is possible to consider alternative losses to induce different optimization behavior, such as in Li and Turner (2016); Dieng et al. (2017).

## 3 Background

In what follows, we consider a latent variable model with a joint probability $p_\theta(x, z) = p(x|z)p(z)$, where $x$ and $z$ are observed and real-valued unobserved variables, and $\theta$ denotes the set of parameters to be learned. Due to the non-conjugate prior-likelihood pair or the non-linearity of the conditioning, exact inference is often intractable. Direct maximization of the marginal likelihood is impossible because the marginalization involves integration: $\log p(x) = \log \int p(x, z) \, \mathrm{d}z$. Thus, training is usually conducted by maximizing the *expected complete data log likelihood* (ECLL) over an auxiliary distribution $q$:

$$\max_\theta \mathbb{E}_{q(z)}[\log p_\theta(x, z)].$$

When using the conditional $q(z|x)$ we emphasize the use of a recognition network in amortized VI (Kingma and Welling, 2014; Rezende et al., 2014). When the exact posterior $p(z|x)$ is tractable, one could choose to use $q(z) = p(z|x)$, which leads to the well known *Expectation-Maximization* (EM) algorithm. When exact inference is not possible, we need to approximate the true posterior, usually by sampling from a Markov chain in MCMC or from a variational distribution in VI. The approximation induces **bias** in learning the model $\theta$, as

$$\mathbb{E}_q[\log p_\theta(x, z)] = \log p_\theta(x) + \mathbb{E}_q[\log p_\theta(z|x)].$$

That is to say, maximizing ECLL increases the marginal likelihood of the data while biasing the true posterior to be more like the auxiliary distribution. The second effect vanishes when $q(z)$ approximates $p(z|x)$ better.

In this paper we focus on the case of VI, where learning of the auxiliary distribution, i.e. variational distribution, is through maximizing the ELBO, which is equivalent to minimizing $\mathcal{D}_{\mathrm{KL}}(q(z)||p(z|x))$. Due to the *zero-forcing* property of the KL, $q$ tends to be unimodal and more concentrated. We emphasize that a highly flexible parametric form of $q$ can potentially alleviate this problem but does not address the issue of finding the optimal parameters. Before going into technical details of choice of $q$ and loss function, we now discuss a few implications and effects doing approximate inference has in practice:

1. At initialization, the true posterior is likely to be multi-modal. Having a unimodal $q$ helps to regularize the latent space, biasing the true posterior towards being unimodal as well. Depending on the type of data being modeled, this may be a good property to have. A unimodal approximate posterior also means less noise when sampling from $q$. This facilitates learning by having lower variance gradient estimates.

2. In cases where the true posterior has to be multi-modal, biasing the posterior to be unimodal inhibits the model from learning the true generative process of the data. Allowing sufficient

uncertainty in the approximate posterior encourages the learning process to explore the spurious modes in the true posterior.

3. Beta-annealing facilitates point 1 by lowering the penalty of the prior contrastive term, allowing the $q(z)$ estimate to be sharp (per point 1). Alpha-annealing encourages exploration (point 2) by lowering the penalty of the cross-entropy term, increasing the importance of the entropy term, and therefore the tendency to explore the latent space (per point 2).

### 3.1 Assumption of the variational family

Recent works have shown promising results in having more expressive parametric form of the variational distribution. This allows the variational family to cover a wider range of distributions, ideally including the true posterior that we seek to approximate. For instance, the *Hierarchical Variational Inference* (HVI) methods (Ranganath et al., 2016) are a generic family of methods that subsume discrete mixture proposals (e.g. mixture of Gaussian), auxiliary variable methods (Agakov and Barber, 2004; Maaløe et al., 2016), and normalizing flows (Rezende and Mohamed, 2015) as subfamilies.

In HVI, we use a latent variable model $q(z_T) = \int q(z_T, z_{t<T}) \, d \, z_{t<T}$ as approximate posterior, with the index $t < T$ indicating the latent variable [1]. Thus the entropy term $-\mathbb{E}_{q(z_T)}[\log q(z_T)]$ is intractable, and needs to be approximated. One can lower bound this term by introducing a reverse network $r(z_{t<T}|z_T)$:

$$-\mathbb{E}_{q(z_T)}[\log q(z_T)] \geq -\mathbb{E}_{q(z_T)}\left[\log q(z_T) + \mathcal{D}_{\mathrm{KL}}(q(z_{t<T}|z_T)||r(z_{t<T}|z_T))\right]$$
$$= -\mathbb{E}_{q(z_T, z_{t<T})}\left[\log q(z_T|z_{t<T})q(z_{t<T}) - \log r(z_{t<T}|z_T)\right].$$

The variational lower bound then becomes:

$$\mathcal{L}(x) \doteq \mathbb{E}_{q(z_T, z_{t<T})}\left[\log \frac{p(x, z_T)r(z_{t<T}|z_T)}{q(z_T|z_{t<T})q(z_{t<T})}\right]. \tag{3}$$

Since $q(z_T)$ can be seen as an infinite mixture model, in principle it has the capability to represent any posterior distribution given enough capacity in the conditional. In the special case where $q(z_T|z_{t<T})$ is deterministic and invertible, we can choose $r$ to be its inverse function. The KL term would vanish, and the entropy can be computed recursively via the change-of-variable formula: $q(z_t) = q(z_{t-1})|\frac{\partial z_t}{\partial z_{t-1}}|^{-1}$. Universal approximation results of certain parameterizations of invertible functions have also been recently shown in Huang et al. (2018).

Expressive as these methods can be, they may be susceptible to bad local minima when fitting a target distribution with well separated modes. Furthermore, we demonstrate in Section 5.2 and 5.3 (and also Appendix D) that they are not robust to beta-annealing.

### 3.2 Loss function tempering: annealed importance sampling

As explained in the introduction, the purpose of doing alpha-annealing is to let the variational distribution be more exploratory early on during training. *Annealed importance sampling* (AIS) is an MCMC method that encapsulates this concept (Neal, 2001). In AIS, we consider an extended state space with $z_0, z_1, ..., z_T$, where $z_0$ is sampled from a simple distribution (such as the prior distribution $p(z)$), and the subsequent particles are sampled from the transition operators $q_t(z_t|z_{t-1})$. We design a set of intermediate target densities as $\tilde{f}_t = \tilde{f}_T^{\alpha_t} \tilde{f}_0^{1-\alpha_t}$, for $t \in [0, T]$, where $\tilde{f}_t \propto f_t$, $f_T$ is the true posterior we want to approximate, and $f_0$ is the initial distribution $z_0$ is sampled from. These (Markov chain) transitions are constructed to leave the intermediate targets $f_t(z)$ invariant, and the importance weight is calculated as

$$w_j = \frac{\tilde{f}_1(z_1)}{\tilde{f}_0(z_1)} \frac{\tilde{f}_2(z_2)}{\tilde{f}_1(z_2)} ... \frac{\tilde{f}_T(z_T)}{\tilde{f}_{T-1}(z_T)}.$$

A downside of AIS is that it requires constructing a long sequence of transitions in order for the intermediate targets to be close enough and thus for the estimate to be accurate.

# 4 Annealed variational objectives

Inspired by AIS and alpha-annealing, we propose to integrate energy tempering into the optimization objective of the variational distribution. As in AIS, we consider an extended state space with random variables $z_0, ..., z_T$, and we take the marginal $q_T(z_T)$ as approximate posterior. We construct $T$ forward transition operators and $T$ backward operators, and assign an intermediate target $\tilde{f}_t$ to each transition pair, which is defined as an interpolation between the true (unnormalized) posterior and the initial distribution: $\tilde{f}_t = \tilde{f}_T^{\alpha_t} \tilde{f}_0^{1-\alpha_t}$, where $0 = \alpha_0 < \alpha_1 < ... < \alpha_T = 1$ [2]. Different from AIS, we propose to learn the parametric transition operators. More formally, we define:

$$q_t(z_t, z_{t-1}) = q_{t-1}(z_{t-1})q_t(z_t|z_{t-1}), \text{ and } q_t(z_t) = \int_{z_0,...,z_{t-1}} q_t(z_t|z_{t-1}) \prod_{t'=0}^{t-1} q_{t'}(z_{t'}|z_{t'-1}) \, d z_{t'}.$$

Set $q_0(z_0) = f_0(z_t)$ [3], which is easy to sample from. We define the backward transitions in the reverse ordering, i.e. $r_t(z_{t-1}, z_t) = r_t(z_{t-1}|z_t)$. We consider maximizing the following objective, which we refer to as the *Annealed Variational Objectives* (AVO):

$$\max_{q_t(z_t|z_{t-1}), r_t(z_{t-1}|z_t)} \mathbb{E}_{q_t(z_t|z_{t-1})q_{t-1}(z_{t-1})} \left[ \log \frac{\tilde{f}_t(z_t)r_t(z_{t-1}|z_t)}{q_t(z_t|z_{t-1})q_{t-1}(z_{t-1})} \right] \tag{4}$$

for $t \in [1, T]$, by taking the partial derivative with respect to the parameters $\phi_t$ of the $t$'th transition (fixing the intermediate marginal $q_{t-1}(z_{t-1})$) [4].

Intuitively, the goal of each forward transition is to stochastically transform the samples drawn from the previous step into the intermediate target distribution assigned to it. Since each intermediate target only differs from the previous one slightly, each transition only needs to correct the marginal samples from the previous step incrementally. Also, in the case of amortized VI, we set $f_0$ to be the prior distribution of the VAE, so each intermediate target has a more widely distributed density function, which facilitates exploration of the transition operators.

## 4.1 Analysis on the optimality of transition operators

The following proposition shows that if each forward transition and backward transition are locally optimal, the result is globally consistent, which means the marginals $r_t^*(z_t)$ and $q_t^*(z_t)$ constructed by locally optimal transitions match the intermediate target $f_t(z_t)$. The corollary below shows that the optimal transitions in Equation 4 bridge the variational gap.

**Proposition 1.** *Let $q_t^*(z_t|z_{t-1})$ and $r_t^*(z_{t-1}|z_t)$ be the functional solutions to Equation 4. Then the resulting marginal density functions $q_t^*(z_t)$ and $r_t^*(z_t)$ are equal to $f_t(z_t)$.*

**corollary 1.** *Take $f_T(z_T)$ as the true posterior distribution $p(z_T|x)$, or simply choose $\tilde{f}_T = p(x|z_T)p(z_T)$. The variational lower bound in Equation 3 is exact with respect to the marginal likelihood $p(x)$ if optimal transitions $q_t^*$ and $r_t^*$ are used.*

The above results demonstrate that optimizing the transitions according to Equation 4 is consistent with the variational Bayes objective when all of them are optimal with respect to the objectives they are assigned to. The proof can be found in Appendix A.

## 4.2 Specification of the transitions

When $\alpha_{t-1}$ is close enough to $\alpha_t$, assuming the marginal $q_{t-1}(z_{t-1})$ approximates $f_{t-1}$ accurately, the transition $q_t(z_t|z_{t-1})$ only needs to modestly refine the samples from $q_{t-1}(z_{t-1})$. In light of this, we design the following *stochastic refinement operator* for both $q_t(z_t|z_{t-1})$ and $r_t(z_{t-1}|z_t)$:

$$q_t(z_t|z_{t-1}) = \mathcal{N}(z_t|\mu_t^q(z_{t-1}), \sigma_t^q(z_{t-1})); \quad r_t(z_{t-1}|z_t) = \mathcal{N}(z_{t-1}|\mu_t^r(z_t), \sigma_t^r(z_t))$$

where the mean $\mu$ and standard deviation $\sigma$ of the conditional normal distribution is defined as follows

$$\mu(z) = g(z) \odot m(z) + (1 - g(z)) \odot z; \quad \sigma(z) = \text{act}_\sigma(W_\sigma \cdot h(z) + b_\sigma)$$

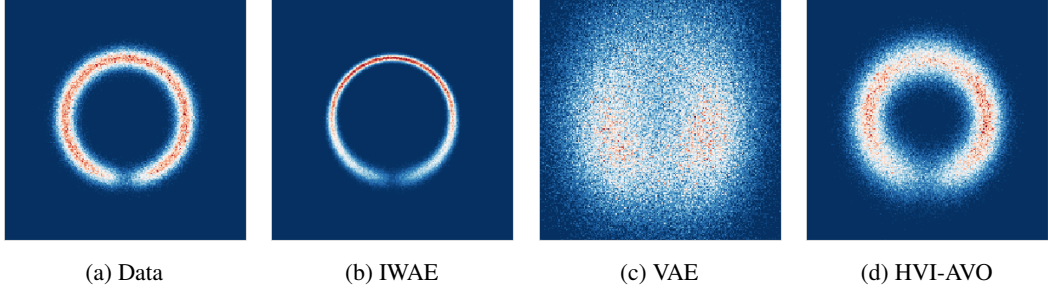

(a) Data      (b) IWAE      (c) VAE      (d) HVI-AVO

Figure 1: Learning the noise process. (a) the true data distribution. (b-d) the distributions learned by the specified methods.

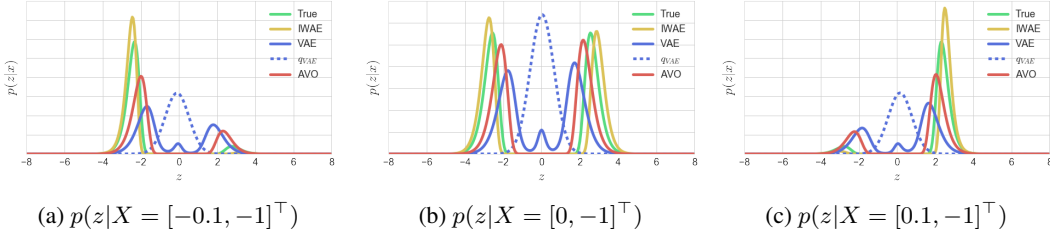

(a) $p(z|X = [-0.1, -1]^\top)$      (b) $p(z|X = [0, -1]^\top)$      (c) $p(z|X = [0.1, -1]^\top)$

Figure 2: Learned posteriors given different locations where inference is ambiguous.

$$m(z) = W_m \cdot h(z) + b_m; \quad g(z) = \text{act}_g(W_g \cdot h(z) + b_g); \quad h(z) = \text{act}_h(W_h \cdot z + b_h).$$

The activation functions $\text{act}_\sigma$, $\text{act}_g$ and $\text{act}_h$ are chosen to be softplus, sigmoid and ReLU (Nair and Hinton, 2010) (or ELU (Clevert et al., 2016) in the case of amortized VI). In the case of amortized VI, we replace the dot product with the conditional weight normalization (Salimans and Kingma, 2016) operator proposed in Krueger et al. (2017); Huang et al. (2018).

### 4.3 Loss calibrated AVO

Since the correctness of the marginal $q_T(z_T)$ trained with AVO depends on the optimality of each transition operator, when used for amortized VI each update will not necessarily improve the marginal to be a better approximate posterior. Hence, in amortized VI, we consider the following loss calibrated version of AVO [5]:

$$\max_{q_t(z_t|z_{t-1}), r_t(z_{t-1}|z_t)} a \, \mathbb{E}_{q_t(z_t|z_{t-1})q_{t-1}(z_{t-1})} \left[ \log \frac{\tilde{f}_t(z_t) r_t(z_{t-1}|z_t)}{q(z_t|z_{t-1}) q(z_{t-1})} \right] + (1-a) \, \mathcal{L}(x)$$

for all $t$, where $a \in [0, 1]$ is a hyperparameter that determines the weight of AVO used in training. In practice naive implementation of this objective can be computationally expensive, so we select the loss function stochastically (with probability $a$ maximize AVO, otherwise maximize ELBO) in the amortized VI experiments, which we found helps to make progress in improving the model $p_\theta(x, z)$.

## 5 Experiments

Our first experiment shows that having a unimodal approximate posterior is not universally benign. The second and third experiments analyze the effect of the optimization process has on the learned (marginal) density of $q_T$, qualitatively and quantitatively. We do this in an unamortized setup, in an attempt to suppress the confounding effect of suboptimal amortization. Finally, we experiment with real data and demonstrate that HVI can be improved with our proposed method.

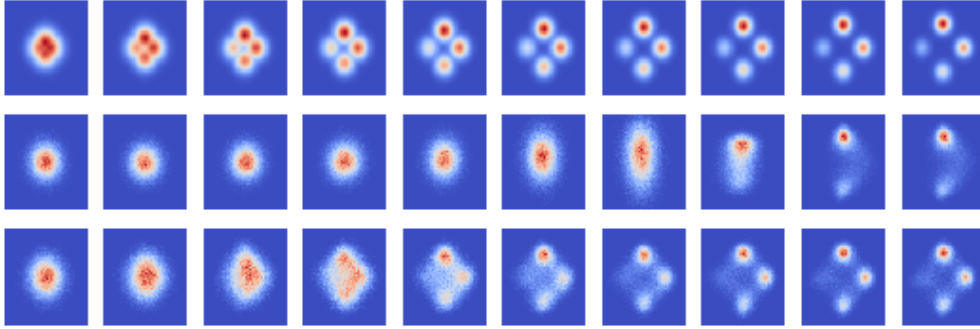

Figure 3: Learned marginals $q_t(z_t)$ at different layers ($T = 10$). First row: true targets $f_1...f_{10}$. Second row: HVI-ELBO. Third row: HVI-AVO.

## 5.1 Biased noise model

To demonstrate the biasing effect of the approximate posterior in amortized VI, we utilize a toy example in the form of the following data distribution:

$$z \sim \mathcal{N}(0, 1), \qquad \epsilon_1, \epsilon_2 \sim \mathcal{N}(0, \varepsilon), \tag{5}$$

$$x_1 = \sin\left(\pi \tanh\left(\eta z\right)\right) + \epsilon_1, \quad x_2 = \cos\left(\pi \tanh\left(\eta z\right)\right) + \epsilon_2, \tag{6}$$

where $\eta = 0.75$. The result is the data distribution depicted in Figure 1a. The data distribution is constructed so that $x = [0, -1]^\top$ has a bimodal true posterior with a mode at the tail ends of the Gaussian. This is analogous to some real data distribution in which two data points close in the input space can come from two well separated modes in latent space (e.g. someone wearing glasses vs. not wearing glasses).

We train three different models which differ only in the way their approximate posterior is parameterized: (1) $q(z|x) = p(z)$, trained using the IWAE loss and 500 samples, (2) A VAE model with a Gaussian approximate posterior $q(z|x) = \mathcal{N}(z|x)$, and finally (3) A VAE model with an AVO loss. We use a decoder *exactly* as in Equation 6 fixed for the mean of the conditional Gaussian likelihood, with standard deviation of $\epsilon_1$ and $\epsilon_2$ parameterized as an MLP conditioned on $z$ to be learned. The densities learned by each model is depicted in Figure 1. We estimate the posterior of the learned generative model in each case, and compare them in Figure 2. At the point $x = [0, -1]^\top$, the true posterior is a bimodal distribution as discussed before. Shifting slightly to the left or right in $x$-space towards the higher density regions gives a sharper prediction in the latent space at their respective ends. In the unambiguous regions of the data, the posterior is unimodal.

We observe that the VAE encoder predicts the posterior to be centered around $z = 0$. From Figure 2 we can clearly see the zero-forcing effect mentioned before, where encoder matches one of the modes of the true posterior. As a consequence of that behavior, the variance of the decoder's prediction at that point is high, resulting in the plot seen in Figure 1c. The IWAE model on the other hand, matches the distribution better. With 500 samples per data point in this case, we are effectively training the generative model with a non-parametric posterior that can perfectly fit the true posterior. The price of doing so, however, is that we require many samples in order to train the model.

For a simple task such as this one, this approach works, but is meant to demonstrate the benefits of having sufficient uncertainty in the proposal distribution. The proposed AVO method ($T = 10$) also approximates the distribution well, but without incurring the same computational cost as IWAE.

## 5.2 Toy energy fitting

In this experiment, we take a four-mode mixture of Gaussian as a true target energy, and compare HVI trained with ELBO and HVI trained with AVO (with $T = 10$). In Figure 3 we see that HVI-ELBO overfits two modes out of four, whereas HVI-AVO captures all four modes. Also, first few layers of HVI-ELBO do not contribute much to the resulting marginal $q_T(z_T)$, where each layer of HVI-AVO has to fit the assigned target.

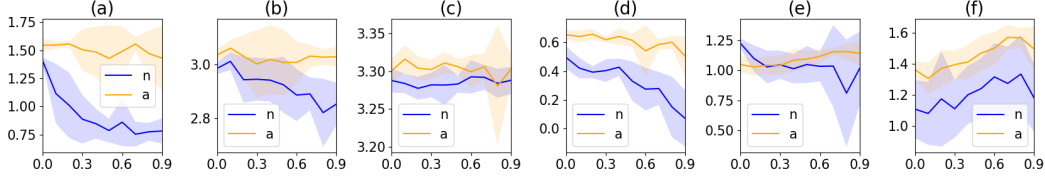

Figure 4: Robustness of HVI to beta-annealing. "n" denotes normal ELBO and "a" denotes AVO. x-axis: the percentage of total training time it takes to anneal $\beta$ linearly back to 1; y-axis: estimated $\mathbb{E}_q\left[-\mathcal{E}(x) - \log q(x)\right]$ (negative KL) shifted by a log normalizing constant (the higher the better).

Table 1: Amortized VI with VAE. $\mathcal{L}_*$ is the ELBO, with $tr$, $va$ and $te$ representing the training, validation and test set respectively. $\log p(\mathcal{D}_*)$ is the estimated log-likelihood on the dataset. $L$ indicates the number of leap-frog steps used in Salimans et al. (2015).

<table>
<tr><td></td><td colspan="6">(a) MNIST</td><td colspan="3">(b) OMNIGLOT</td></tr>
<tr><td></td><td>$T=0$<br>$a=0$</td><td>$T=5$<br>$a=0$</td><td>$T=5$<br>$a=0.2$</td><td>$T=5$<br>$a=0.5$</td><td>$L=4$</td><td>$L=8$</td><td>$T=0$<br>$a=0$</td><td>$T=5$<br>$a=0$</td><td>$T=5$<br>$a=0.5$</td></tr>
<tr><td>$-\mathcal{L}_{tr}$</td><td>84.69</td><td>84.51</td><td>86.15</td><td>87.02</td><td>-</td><td>-</td><td>106.67</td><td>111.32</td><td>108.94</td></tr>
<tr><td>$-\mathcal{L}_{va}$</td><td>92.37</td><td>92.37</td><td>90.00</td><td>90.22</td><td>-</td><td>-</td><td>114.68</td><td>110.78</td><td>109.47</td></tr>
<tr><td>$-\log p(\mathcal{D}_{va})$</td><td>87.49</td><td>87.63</td><td>86.06</td><td>86.12</td><td>-</td><td>-</td><td>108.41</td><td>106.32</td><td>105.18</td></tr>
<tr><td>$-\mathcal{L}_{te}$</td><td>92.40</td><td>92.48</td><td>90.01</td><td>90.26</td><td>89.82</td><td>88.3</td><td>115.25</td><td>113.19</td><td>111.94</td></tr>
<tr><td>$-\log p(\mathcal{D}_{te})$</td><td>87.50</td><td>87.62</td><td>86.06</td><td>86.12</td><td>86.40</td><td>85.51</td><td>108.55</td><td>106.25</td><td>105.04</td></tr>
<tr><td>gen gap</td><td>7.71</td><td>7.97</td><td>3.86</td><td>3.24</td><td>-</td><td>-</td><td>8.58</td><td>1.87</td><td>3.00</td></tr>
<tr><td>var gap</td><td>4.90</td><td>4.86</td><td>3.95</td><td>4.14</td><td>3.42</td><td>2.79</td><td>6.70</td><td>6.94</td><td>6.90</td></tr>
</table>

## 5.3 Quantitative analysis on robustness to beta annealing

We perform the experiments in Section 5.2 again, on six different energy functions, $\mathcal{E}$ (summarized in Appendix B) with different linear schedules of beta-annealing. We run 10 trials for each energy function, and do 2000 stochastic updates with a batch size of 64 and a learning rate of 0.001. We evaluate the resulting marginal $q_T(z_T)$ according to its negative KL divergence from the true energy function. Since the entropy of $q_T(z_T)$ is intractable, we estimate it using importance sampling; we elaborate this in Appendix C. We see that HVI-AVO constantly outperforms HVI-ELBO. Even when most of the time performance of HVI-ELBO degrades along with prolonged beta-annealing schedule, HVI-AVO's performance is relatively invariant. In Appendix D, we visualize the learned marginals $q_t(z_t)$ under beta annealing using both stochastic and deterministic transitions.

## 5.4 Amortized inference

We train VAEs using a standard VAE (with a Gaussian approximate posterior), HVI, and HVI with AVO on the Binarized MNIST dataset from Larochelle and Murray (2011), and the Omniglot dataset as used in Burda et al. (2016). In these experiments, we used an MLP for both the encoder and decoder, with gating activation as in Tomczak and Welling (2016). Both the encoder and decoder have 2 hidden layers, each with 300 hidden units. For MNIST, we used a dimension of 40 for the latent space, and 200 dimensions in Omniglot. We used hyperparameter search to determine batch size, learning rate, and the beta-annealing schedule.

Table 1a lists the results from our experiments on MNIST, alongside results from Salimans et al. (2015) combining MCMC in HVI for comparison (see Appendix E). Table 1b lists the results for Omniglot. We find that HVI's approximate posterior trained with AVO tends to have smaller variational gap (estimated by difference between test likelihood and ELBO), and also better test likelihood. It is also worth noting that in the MNIST case, smaller variational gap translates into smaller generalization gap and also better test likelihood, while the same is not true in the OMNIGLOT example, which corroborates our finding in Section 5.1 that true posterior can be biased towards approximate posterior, resulting in a smaller variational gap (8.58) but a worse density model (6.70).

# 6 Conclusion

We find that despite the representational capacity of the chosen family of approximate distributions in VI, the density that can be represented is still limited by the optimization process. We resolve this by incorporating annealed objectives into the training of hierarchical variational methods. Experimentally, we demonstrate (1) our method's robustness to deterministic warm-up, (2) the benefits of encouraging exploration and (3) the downside of biasing the true posterior to be unimodal. Our method is orthogonal to finding a more rich family of variational distributions, and sheds light on an important optimization issue that has thus far been neglected in the amortized VI literature.

# 7 Acknowledgements

We would like to thank Massimo Cassia and Aristide Baratin for their useful feedback and discussion.

## Footnotes

[1] Here we consider sequence of latent variables $z_t$ of the same dimension as $z_T$, but in principle the dimensionality can vary.

[2] In the experiments, we use a linear annealing schedule for all models trained with AVO.

[3] We learn the initial distribution $q_0(z_0|x)$ in the case of amortized VI.

[4] In practice, this is achieved by disconnecting the gradient.

[5] Note that in amortized VI $\tilde{f}_t$, $r_t$ and $q_t$ all depend on the input $x$, but we omit the notation for simplicity.

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
