[Supplementary Material]

## A    Proof of optimal transitions

This is the proof of Proposition 1.

*Proof.* The base case when $t = 0$ is trivially true by choice of initial distribution. We now look at the case when $t > 0$. Taking the *functional derivative* of the objective defined by Equation 4, along with a Lagrange multiplier to satisfy the second axiom of probability (both $q$ and $r$ sum to one), with respect to the conditionals $q$ and $r$ yields:

$$q_t^*(z_t|z_{t-1}) = \frac{f_t(z_t)r_t^*(z_{t-1}|z_t)}{Z_t^q(z_{t-1})}; \quad r_t^*(z_{t-1}|z_t) = \frac{f_{t-1}(z_{t-1})q_t^*(z_t|z_{t-1})}{Z_t^r(z_t)} \tag{7}$$

where we take $q(z_{t-1}) = f_{t-1}(z_{t-1})$, assuming it's optimal, and $Z(\cdot)$'s are normalizing constant given the argument, and are proper distributions since the numerators above are both proper joint probabilities. Taking the product of the two equations yields

$$\frac{f_t(z_t)f_{t-1}(z_{t-1})}{Z_t^r(z_t)Z_t^q(z_{t-1})} = 1$$

Thus $Z_t^r(z_t) = f_t(z_t)$ and $Z_t^q(z_{t-1}) = f_{t-1}(z_{t-1})$ by marginalization. By definition and the induction assumption, the resulting mariginal of the optimal forward transition is:

$$q_t^*(z_t) = \int_{z_{t-1}} f_{t-1}(z_{t-1})q_t^*(z_t|z_{t-1}) \,\mathrm{d}\, z_{t-1}$$

$$= \int_{z_{t-1}} r_t^*(z_{t-1}|z_t)Z_t^r(z_t) \,\mathrm{d}\, z_{t-1} = f_t(z_t)$$

Similarly, the resulting marginal of the optimal forward transition is:

$$r_{t-1}^*(z_{t-1}) = \int_{z_t} f_t(z_t)r_t^*(z_{t-1}|z_t) \,\mathrm{d}\, z_t$$

$$= \int_{z_t} q_t^*(z_t|z_{t-1})Z_t^q(z_{t-1}) \,\mathrm{d}\, z_t = f_{t-1}(z_{t-1})$$

Since both the base case and inductive step are true, by mathematical induction, $q_t^*(z_t) = r_t^*(z_t) = f(z_t)$ for all $t \in [0, T]$. □

This is the proof of Corollary 1.

*Proof.* Due to Equation 7, we can compute the optimal transition ratio as $\frac{r_t^*(z_{t-1}|z_t)}{q_t^*(z_t|z_{t-1})} = \frac{f_{t-1}(z_{t-1})}{f_t(z_t)}$. We can expand Equation 3 as follows:

$$\mathbb{E}_{q(z_T...z_0)} \left[ \log \frac{p(x, z_T)r_T^*(z_{T-1}|z_T)...r_{T-1}^*(z_0|z_1)}{q_T^*(z_T|z_{T-1})q_{T-1}^*(z_{T-1}|z_{T-2})...q_0(z_0)} \right]$$

$$= \mathbb{E}_{q^*(z_T...z_0)} \left[ \log p(x, z_T) + \frac{r_T^*(z_{T-1}|z_T)}{q_T^*(z_T|z_{T-1})} + ... + \frac{r_1^*(z_0|z_1)}{q_1^*(z_1|z_0)} - \log q_0(z_0) \right]$$

$$= \mathbb{E}_{q^*(z_T...z_0)} \left[ \log p(x, z_T) + \frac{f_{T-1}(z_{T-1})}{f_T(z_T)} + ... + \frac{f_0(z_0)}{f_1(z_1)} - \log q_0(z_0) \right]$$

$$= \mathbb{E}_{q^*(z_T...z_0)} \left[ \log \frac{p(x, z_T)f_{T-1}(z_{T-1})...f_0(z_0)}{f_T(z_T)f_{T-1}(z_{T-1})...q_0(z_0)} \right] = \mathbb{E} \left[ \log \frac{p(x, z_T)}{p(z_T|x)} \right] = \log p(x)$$

since $f_T(z_T) = p(z_T|x)$ and $q_0(z_0) = f_0(z_0)$. □

# B Specification of the toy energy functions

Visualization of the toy energy functions, $\mathcal{E}$, are presented in Figure 5. Below are the corresponding formulas.

Figure 5: From left to right top to bottom, the energy functions (a) to (f)

Table 2: Formulas of toy energy functions

| | |
|---|---|
| (a) | $-\frac{((\|z\|_2-2)/0.4)^2}{2} + \log\left(\exp\left(-\frac{((z_1-2)/0.6)^2}{2}\right) + \exp\left(-\frac{((z_1+2)/0.6)^2}{2}\right)\right)$ |
| (b) | $-\frac{((\|0.5z\|_2-2)/0.5)^2}{2} + \log\left(\exp\left(-\frac{((z_1-2)/0.6)^2}{2}\right) + \exp\left(-\frac{(2\sin z_1)^2}{2}\right) + \exp\left(-\frac{((z_1+z_2+2.5)/0.6)^2}{2}\right)\right)$ |
| (c) | $-\left(2 - \sqrt{z_1^2 + \frac{z_2^2}{2}}\right)^2$ |
| (d) | $\log\left(0.1\mathcal{N}([-2,0]^\top, 0.2I) + 0.3\mathcal{N}([2,0]^\top, 0.2I) + 0.4\mathcal{N}([0,2]^\top, 0.2I) + 0.2\mathcal{N}([0,-2]^\top, 0.2I)\right)$ |
| (e) | $-\frac{((z_2-w_1)/0.4)^2}{2} - 0.1(z_1^2)$ |
| (f) | $\log\left(\exp\left(-\frac{((z_2-w_1)/0.35)^2}{2}\right) + \exp\left(-\frac{((z_2-w_1+w_2)/0.35)^2}{2}\right)\right) - 0.05z_1^2$ |
| | $w_1 = \sin\left(\frac{\pi}{2}z_1\right) \qquad w_2 = 3\exp\left(-\frac{(z_1-2)^2}{2}\right)$ |

# C Estimating $\mathcal{D}_{\mathrm{KL}}(q_T(z_T)\|f_T(z_T))$

Similar to Burda et al. (2016) we upper bound the negative KL:

$$\mathbb{E}_{q_T(z_T)}[\log f_T(z_T) - \log q_T(z_T)]$$

$$= \mathbb{E}_{q_T(z_T)}[\log f_T(z_T) - \log \int_{z_{<T}} q_T(z_T|z_{<T})q(z_{<T})\,\mathrm{d}z_{<T}]$$

$$= \mathbb{E}[\log f_T(z_T) - \log \int_{z_{<T}} r(z_{<T}|z_T)\frac{q_T(z_T|z_{<T})q(z_{<T})}{r_T(z_{<T}|z_T)}\,\mathrm{d}z_{<T}]$$

$$\leq \mathbb{E}_{q_T(z_T)}[\log f_T(z_T) - \mathbb{E}_{r_T(z_{<T}|z_T)}[\log \sum_{i=1}^{K} \frac{1}{K}\frac{q_T(z_T|z_{<T,i})q_T(z_{<T,i})}{r_T(z_{<T,i}|z_T)}]]$$

Moreover, if $w = \frac{q_T(z)}{r_T(z_{<T}|z_T)}$ is bounded, then by the *strong law of large number*, $\frac{1}{K}\sum_{i=1}^{K} w_i$ converges almost surely to $\mathbb{E}_r[\frac{q_T(z_T,z_{<T})}{r(z_{<T}|z_T)}] = q_T(z_T)$. That is, the gap is closed up by using large number of samples of $z_{<T}$ generated by $r_T(z_{<T}|z)$. In our case, $T = 5$ or $10$, so both $q_T(z_T|z_{<T})$ and $r_T(z_{<T}|Z_T)$ further factorize as product of conditional transitions. The bound is tight with around 100 samples in our examples. We use 2000 samples simply to reduce variance to estimate the KL to evaluate the learned $q_T(z_T)$.

# D    Visualization of robustness to beta-annealing

Figure 6: Learned marginals $q_t(z_t)$ at different layers ($T = 10$) trained with beta-annealing. First row: HVI-ELBO. Second row: HVI-AVO.

We further simulate the scenario of VAE training in which beta-annealing of the objective is applied to encourage the approximate posterior to be more deterministic. Since there is no concept of prior distribution in the case of energy function, we only decrease the weight of all entropy and cross-entropy terms of the transition operators. The annealing coefficient $\beta$ is initially set to be 0.01 and annealed back to 1.0 linearly throughout 80% of the training time (2500 updates). We see in Figure 7 that HVI-ELBO collapses to only one mode, and HVI-AVO remains robust even in the face of an annealing scheme that discourages exploration.

Figure 7: Learned marginals $q_t(z_t)$ at different layers ($T = 10$) trained with beta-annealing. First row: DSF-ELBO. Second row: DSF-AVO.

The same can be found with deterministic transitions, i.e. normalizing flows. We experimented with the Deep Sigmoidal Flows, a universal change of variable model proposed by Huang et al. (2018), and found that problem of lack of exploration was worsened (hypothetically due to lack of noise injection in the transition operators) and can be remedied by AVO as well.

# E    Discussion on Table 1

It is important to note that the results in Salimans et al. (2015) are not directly comparable to ours due to different architectures. The architecture we used was from Tomczak and Welling (2016), who provide a baseline for using Householder flow, with an estimate of 87.68 on negative log likelihood (NLL) using 5000 samples for importance weighted lower bound (whereas we use 1000). Tomczak and Welling (2017) further experimented with inverse autoregressive flow (IAF) with an estimate of 86.70 on NLL, and convex combination of IAFs yield an NLL of 86.10. We showed that our AVO can improve the performance of HVI from 87.62, similar to 21 Householder flow, to 86.06, which is better than all the abovementioned flow based methods applied to the same encoder and decoder architecture.