[Reviews · NeurIPS 2018]

Reviewer 1



Summary: motivated from annealed importance sampling (AIS), the paper introduces a method for variational inference that specifies a sequence of annealed posterior targets f_t(z) for the marginal variational posterior q_t(z_t). These marginals are given by distributions q(z_t|z_(t-1)) defined on an extended state space of latents z_0,…,z_T. The paper shows that in making these marginal variational posteriors approach each target, this eases the optimisation of the q(z_T) towards the true posterior, and helps it explore the various modes of the true posterior, as does alpha-annealing. The method is novel, well-motivated and justified, with a nice overview of the related work in amortised variational inference and AIS. The paper reads well, and shows good technical quality - the AVO is well justified in section 4.1 with Proposition/Corollary 1. My main concern for the method is its practicality. Firstly to evaluate the AVO for each t, one needs to evaluate p(x,z_t), hence a full pass through the decoder is necessary. I imagine this will make the learning quite slow, as is the case with Hamiltonian VI in Salimans et al, where they need one pass through the decoder per leapfrog step (I think this is why they use T=1 in their experiments). It would be good to know how these two compare in running time, when you use the transitions specified in section 4.2 for AVO. Another point that should also be made clear is that these transitions will not leave the intermediate posteriors invariant, which is required for AIS to be valid. If instead you use Hamiltonian dynamics for these transitions (without accept-reject step to keep the procedure differentiable), I imagine the learning would be a lot slower, so although you say the two are orthogonal, it seems that using both would be impractical. Also the marginal log likelihoods in Table 1 appear to be similar if not worse compared to Hamiltonian VI. It would be interesting to have comparisons for OMNIGLOT as well. Also if I understood correctly, AVO is not applicable to deterministic transitions e.g. Normalizing flows, Inverse autoregressive flows, RNVP etc. These all tend to give better likelihoods than those in Table 1 with much faster training (also with fully connected architectures), and are capable of modelling multimodal posteriors, so in what cases I use AVO over these methods? The paper seems to claim in lines 144-145 that these methods are not robust to beta-annealing, but AVO is only compared to Hierarchical VI in Section 5.3. Other minor comments: - line 120: you mean “discourage” instead of “encourage”? - What were the different beta-annealing schedules you tried in section 5.3? - line 225: “Appendix D” the supplementary materials only seem to have up to Appendix B? - Figure 4: On the y-axis, how does the negative KL reach positive values? - Section 5.4 is titled Amortized inference, but for all previous experiments in sections 5.1-5.3 you were also using amortized inference? Overall, the paper is technically sound, novel and reads well, but its significance appears rather limited.

Reviewer 2



Those interested in learning deep latent variable models often rely on heuristics to escape local minima found in parameter space. Two such heuristics are: * alpha annealing -- annealing the log joint of the generative model from 0->1 in the variational bound * beta annealing -- annealing the KL divergence from 0->1 This work aims to incorporate annealing into the variational objective. To achieve this, the work extends an idea from Hierarchical VI. Hierarchical VI uses priors over variational parameters to increase the richness of the variational posterior distribution. Here, the operation to define a prior is used recursively T times to define a transition (and reverse transition so that the entropy can be bounded) from some base distribution. This results in a final variational distribution defined as the marginalization over the entire chain. The idea, while interesting, is similar to that used in the Salimans et. al work (which uses auxillary momentum variables and Hamiltonian dynamics) to iteratively drive changes to the latent state. Here the intuition for the approach is derived from annealed importance sampling. At each step, the goal is to approximate a geometric interpolation between the prior and posterior density. By learning t different parametric forward and backward transition operators (independently, since the marginals of the previous time step are assumed to be fixed), the paper hopes to "learn to anneal" from the prior density to the marginal of the posterior via T intermediate distributions. Training is done using a convex combination of (1) the variational bound and (2) a (sum of?) bound for the marginal likelihood of the intermediate density at each point along the transformation. Experimentally, on synthetic data, learning with AVO is found to learn better generative models than maximizing the ELBO. Compared to IWAE, the method is less computationally intensive. When modeling binarized MNIST HVI [trained with AVO] is found to have better test log-likelihood and lower variational gap (suggesting a learned model for which the inference network provides a closer fit to the generative model's true posterior distribution) compared to learning using the ELBO only. The method however, is outperformed by the Salimans et. al baseline. No explanation is provided for why this might be, though my suspicion is that the gradients from the generative model allow HMC to find good estimates of the posterior density faster. The main strength of this paper is the introduction of an interesting new idea that interprets annealing as part of the variational objective. I was able to follow the paper though for a reader less familiar with all the different ways to write the ELBO, it would prove a challenging read. The readability and clarity of the work have scope for improvement. From a technical standpoint, the work does not see too much improvement in learning compared to the work by Salimans et. al. I have the following clarifying questions about the paper: * What is used for \tilde{f} in the final loss (Section 4.3)? * Is it necessary to hold fixed q_t(z_{t-1})? What is the rationale for not backpropagating the loss through it? * Line 153: phi_t is not previously introduced. Does it represent the parameters of both q and r? * Section 4.2, Where does beta_{t-1} come from? * In both the synthetic examples (examples shown in Figure 1, 3), how do VAEs trained with alpha/beta annealing perform respectively? * How should "a" (in Section 4.3 -- training loss) be selected? * Is there a typo in Proposition 1, should it be r_t*(z_{t-1}|z_t)? * Appendix A, line 346 -- which "objective above"?

Reviewer 3



POST-REBUTTAL Having read the rebuttal, I am inclined to keep my score as is. I like the paper and the main reason why I am not giving a higher score is because I think the proposed solution AVO (an ad hoc collection of objectives) seems rather crude. I look forward to future work that improves upon the currently proposed AVO. I still think that one needs to be careful with the "optimization process" claim. In the rebuttal, the authors stated that "both HVI and NF are highly expressive families of distribution that can provide a good approximate of a potentially highly multimodal target." There are two problems with this line of argument: 1. The correctness of the aforementioned quote depends on the expressivity of amortization family (aka encoder size). In the limit of universal function approximation, I agree with the statement. For finite-depth networks used in practice, I do not think we have the necessarily empirical evidence to back up the claim. 2. Let's suppose the encoder has "infinite capacity." Any argument that there is an issue with the optimization process is *very difficult to support*. If one readily admits that the optimization process is suboptimal, It is difficult to say for sure that the suboptimal optimization process will necessarily result in mode-seeking behavior---even if the objective itself is known to be mode-seeking. This is because the mode-seeking characterization of the objective is only guaranteed for the optimal solution. Ultimately, this issued is resolved via their empirical experiments, which showed consistent mode-collapsing behavior. I therefore think the authors could be more careful about the way they structured their claim. -------------------------------------------------------------------------------- SUMMARY OF PAPER The paper makes the interesting observation that ELBO objective encourages the variational posterior to adopt a mode-seeking behavior, and that this bias will persist even when a more expressive variational family is used due to optimization issues. To remedy this, they introduce a novel regularizer inspired by Annealed Importance Sampling that supposedly biases the variational posterior to be less mode-seeking. They demonstrate the value of the proposed regularizer on several illuminating toy examples as well as MNIST/OMNIGLOT density estimation. -------------------------------------------------------------------------------- SUMMARY OF EVALUATION This is quite an interesting paper that highlights the mode-seeking behavior of variational inference. While the mode-seeking behavior of variational inference is well-known, little has thus far been written in the VAE community about how this biases the generative model to adopt unimodal posteriors. I think this paper will help draw attention to this important issue. Pros: + Generally well-written + Addresses an important problem + Proposes an interesting solution (AVO) + Insightful experiments Cons: - AVO is kind of hacky - Not completely clear why AVO encourages exploration -------------------------------------------------------------------------------- NARRATIVE The narrative is sound; there are only a few things I want to remark: 1. Is the problem the optimization *process* or the optimization objective? To me, optimization process suggests that the problem has to do with the choice of optimizer, and that access to an oracle optimizer will solve the issue. The issue as presented in this paper, however, is regarding the inherent mode-seeking behavior of the variational inference objective which trains the variational posterior via reverse-mode KL. 2. More discussion about the AVO There are a couple of things that make AVO peculiar and worthy of more exposition. First, it is interesting that AVO is not a single optimization problem, but a heuristic collection of T separate optimization problems. My guess is that this design is forced by the fact that optimization of the normalized f-tilde (w.r.t. theta) and the marginal q(z) are intractable. The fact that the AVO is an ad-hoc collection of T objective problems makes it harder to truly characterize how AVO will behave. It also raises the question how the behavior of AVO might have differed had it been a single, fully-differentiable objective trained on both inference and generative model parameters. I think these issues should be discussed in the paper so that future readers are inspired to come up with potentially better solutions than AVO. Second, it is not obvious why AVO necessarily encourages exploration. I would appreciate a more thorough discussion on why the authors believe exploration is implied by AVO. At a very high level, OVA still more or less minimizes the KL(q || f*r), which still (arguably) biases q toward mode-seeking (w.r.t. f*r). Why then is this not a source of concern? On a related note, I think the sentence "in the case of amortized VI, we set f0 to be the prior distribution of the VAE, so each intermediate target has a more widely distributed density function, which facilitates exploration of the transition operators" should be expanded upon to emphasize the potential importance of designing f0 to be a broad distribution. In fact, I would appreciate additional experiments for Table 1 showing what would happen if f0 (and q0) is instead trainable. 3. Checking the unimodality claim The unimodality-bias of ELBO popped up a number of times. While I agree that this is probably true, it would be nice to empirically check whether the ELBO-trained VAEs tend to end up with unimodal posteriors, and whether AVO-trained VAEs end up with non-unimodal posteriors (or something else?). If there is anyway of checking the number of modes in either the true posterior or variational posterior, this would be an interesting experiment. -------------------------------------------------------------------------------- EXPERIMENTS 1. Show ELBO values for Figure 3 I would like to see the ELBO values for the q learned by HVI-AVO v. HVI-ELBO. Ideally, we should see that HVI-AVO has a worse ELBO. 2. Where is appendix D? I only see appendix A and B in the supplementary file. I would like to check how the entropy of q_T(z_T) is computed to get a sense of how tightly the lower bound approximates the the log marginal. 3. Beta-annealing schedule used in 5.3 Can you describe more precisely what is the experimental set-up for 5.3? I'm afraid I don't understand the sentence "x-axis: the percentage of total training time it takes to anneal β back to 1." Is it stated anywhere that the beta-annealing schedule is also linear? 4. AVO applied to amortized VI There are a number of ways one could potentially construct the amortized forward and backward transition operators depending on how you choose to condition on x. It would be nice if graphical model figure is provided somewhere in the paper, showing the exact choices of generative/inference model structure used in the experiments (e.g. see Auxiliary Deep Generative Model Fig. 1 for example). 5. Log-likelihood estimation for Table 1 Please state in the paper how the log-likelihood is estimated. 6. Not setting f0 to be the prior I'm quite interested in what would happen if f0 is not set to be the prior distribution of the VAE.